# Analysis of disaster-affected population mobility through grid-aggregated mobile location data: The 2017 Jiuzhaigou earthquake, China

Zezhi Lin[1,2], Rui Mao[1]*, Huaiqun Zhao[1], Zihui Tang[3], Saini Yang[1], Po Pan[4]

1 School of National Safety and Emergency Management, Beijing Normal University, Beijing, China, 2 School of Systems Science, Beijing Normal University, Beijing, China, 3 School of Architecture and Urban Planning, Beijing University of Civil Engineering and Architecture, Beijing, China, 4 China United Network Communications Co., Ltd. Beijing Company, Beijing, China

* mr@bnu.edu.cn

## Abstract

In disaster research, individual-level mobile phone location data is considered highly valuable for assessing population mobility and disaster impacts. However, due to privacy regulations in China, only spatially aggregated mobile data with a resolution of 1 km × 1 km are available. These data do not contain explicit population individual population movement, which poses challenges for analyzing population movement patterns in disaster research. To using this grid-based mobile data to describe population movement, we applied an empirical orthogonal function (EOF) method to the post-disaster phase of the 2017 Jiuzhaigou earthquake. The first EOF mode (EOF1) primarily exhibits positive anomalies centered over the Jiuzhaigou Valley. The principal components for the EOF1 show a decreasing trend from midnight to 20:00, indicating a continuous outflow of population from the Jiuzhaigou Valley during this period. The second mode (EOF2) exhibits negative anomalies at the Jiuzhaigou Valley and along the road to the southwest of the Valley, while positive anomalies appear along two roads, i.e., one extending from the Jiuzhaigou Valley to Shuanghe, and the other from the Chuanzhusi Town government square to western Chuanzhusi. The primary components of EOF2 reveal that, from midnight to 10:00, population increased along these two roads while decreasing over the Jiuzhaigou Valley and the road leading southward to the Chuanzhusi Town government square. After 10:00, this population change pattern diminished between 10:00–15:00. Based on the EOF2 results, two evacuation routes were identified: Path 1 extended northwest from the Chuanzhusi Town government square; Path 2 led southeast from Jiuzhaigou Valley through Shuanghe Town. In comparison, the BBAC_I clustering method identifies clusters with similar temporal trends but fails to pinpoint the most affected areas or infer evacuation directions. In contrast, EOF analysis overcomes these limitations by revealing key impact zones and evacuation patterns, even in the absence of trajectory data.

**Data availability statement:** The dataset for this study was originally acquired from China Unicom under a confidentiality agreement. This agreement prohibits public sharing of the data without prior written authorization from China Unicom. Therefore, the authors do not have the legal authority to make the dataset publicly available. Interested researchers may request access to the data by contacting: Dr. Shi Shen (Beijing Normal University) – Email: shens@bnu.edu.cn Strategic Customer Department, China Unicom – Phone: +86 18519518331 Approval is required from China Unicom and may involve additional agreements or ethical review, depending on the nature of the request.

**Funding:** This research was supported by the National Key Research and Development Program (2022YFC3004404). The funders had no role in study design, data collection and analysis, decision to publish, or preparation of the manuscript.

**Competing interests:** The authors have declared that no competing interests exist.

## Introduction

Understanding population movement patterns following disasters is essential for effective emergency response, minimizing loss of life, and optimizing the allocation of emergency resources. Traditional methods for population analysis, such as field surveys, provide detailed demographic information and insights into the motivations behind population movements [1]. However, these methods are limited in disaster scenarios due to their high costs, time-consuming data collection, and low resolution for effective emergency response [2,3]. Remote sensing data offers an alternative. For instance, night-time light satellite imagery is useful for estimating population density [4], but faces limitations in disaster population analysis for their invalidation during daylight. General satellite imagery can support daytime assessments of building damage and potential population displacement, but its utility is often hindered by limited spectral resolution constraints, adverse weather conditions, and lengthy data processing times [5,6], hampering its ability to provide real-time support for rescue operations [7].

Given these limitations, the widespread adoption of smartphones offers a valuable alternative for population analysis in disasters. Mobile phone data, with its high spatio-temporal resolution and wide coverage [8,9], has been instrumental in generating spatial-temporal statistical information on human activities across multiple domains, including disaster scenarios [10–17]. By utilizing detailed, user-based mobile location data, researchers can monitor real-time evacuation patterns and examine how population movement is influenced by various factors under disasters, such as earthquake intensity [18], socioeconomic status [19,20], transportation diversity [21], and regional connectivity [22]. These insights enable a deeper understanding of how individuals respond and make decisions during disasters.

Despite these advantages, detailed user-based mobile phone data are rarely accessible due to strict privacy regulations. As a result, many studies rely on grid-based mobile phone data—aggregated from individual records—which provide population counts within defined spatial grids at specific time intervals. However, this aggregation removes user-specific trajectory information, making it challenging to analyze individual-level population movement patterns. While previous studies [23,24] have used such data to evaluate disaster impacts and regional resilience, they generally unable to capture detailed mobility patterns during crises.

To address this issue, researchers have explored various analytical methods that aim to extract meaningful information from grid-level data. Among these, the Bregman Block Average Co-clustering algorithm with I-divergence (BBAC_I) has emerged as an efficient spatiotemporal clustering technique [25] and has been applied to analyze population changes during disaster events, including the 2017 Jiuzhaigou earthquake [26]. BBAC_I is capable of identifying regions with similar temporal trends in population variation. However, it has limited ability to identify critical impact zones or uncover directional movement flows—especially when applied to anonymized, gridded datasets without trajectory data.

To overcome these limitations, this study explores the use of the Empirical Orthogonal Function (EOF) method, which decomposes spatial-temporal data into

orthogonal modes that reflect dominant patterns of variation. By detecting positive and negative anomalies in EOF components, this method allows for the exploration of population changes over time, enabling preliminary assessments of movement directions and intensities during disasters. For example, during earthquakes, populations often evacuate from high-risk areas to safer regions, resulting in population increases in shelters, decreases in hazardous zones, and fluctuations along evacuation routes. By analyzing these population spatial changes with EOF analysis, it becomes possible to infer evacuation directions and patterns even in the absence of individual trajectory data.

Since both the EOF and BBAC_I methods are capable of capturing spatial-temporal population changes, this study applies both approaches to analyze population movements following the 2017 Jiuzhaigou earthquake. Their results were compared to evaluate the strengths and limitations of each approach in capturing post-disaster population dynamics.

The analytical framework of this study is illustrated in Fig 1.

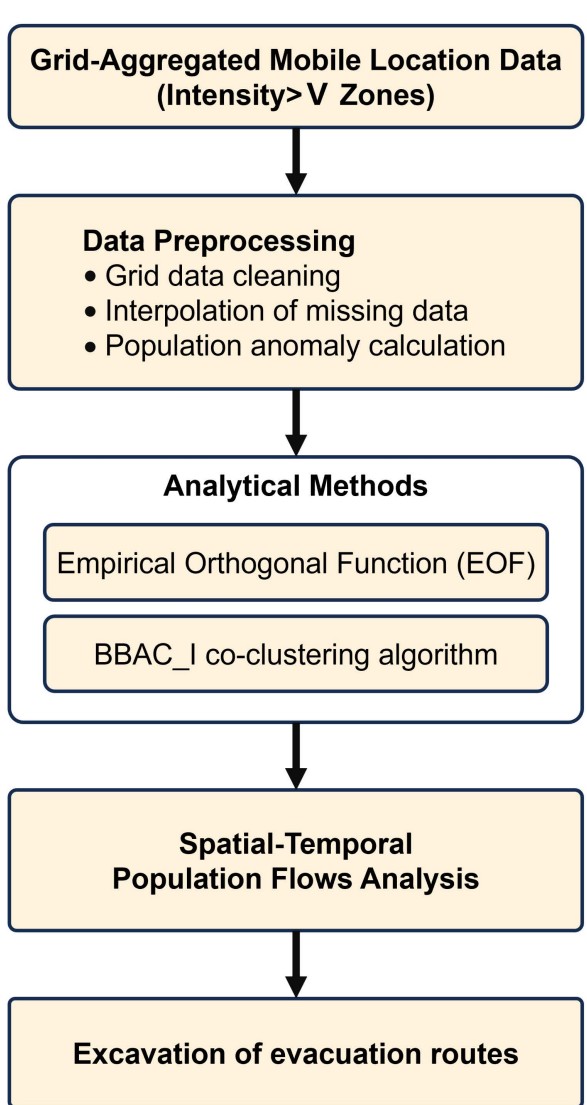

**Fig 1. Analytical framework of the study.**

## Materials and methods

### Data

**Earthquake description and data description.** At 21:19 on August 8, 2017, a magnitude 7.0 earthquake struck Jiuzhaigou County in northern Sichuan Province, China (Fig 2). According to the National Seismological Science Data Center, the epicenter was located at 33.20°N, 103.82°E, with a focal depth of 20 km. This earthquake caused significant casualties and economic losses in the local community. As of 20:00 on August 13, 2017, the earthquake had resulted in 25 deaths, 525 injuries, 6 missing persons, 176,492 people affected, and 73,671 buildings damaged to varying degrees.

The mobile location data used in this study were provided by China Unicom, covering the affected areas of the earthquake with hourly time scale. The dataset includes records from August 2, 2017 (six days before the earthquake), and

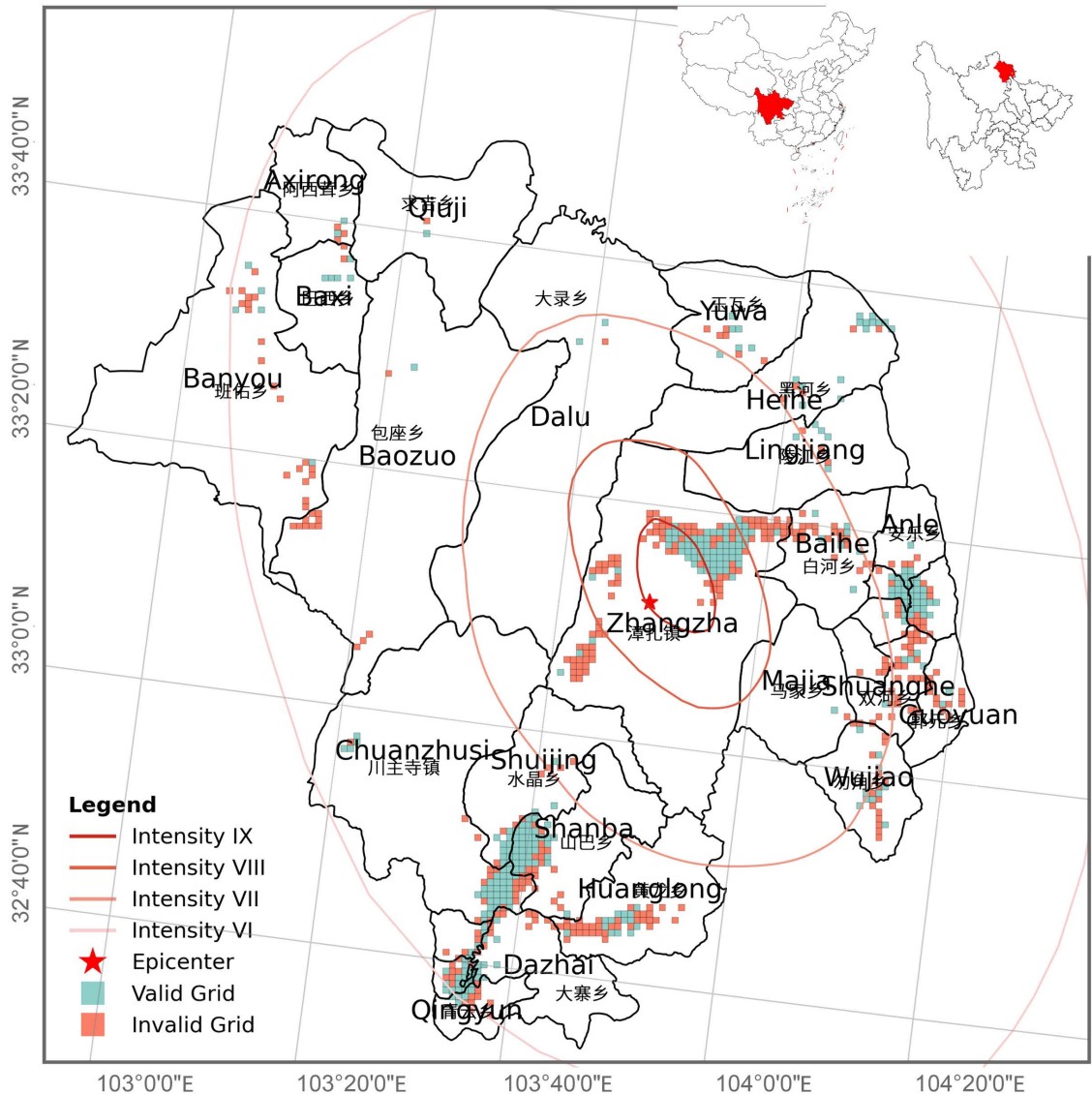

**Fig 2. Spatial intensity distribution on August 9, 2017.**

August 9, 2017 (the day after the earthquake), which were used to represent pre- and post-disaster population distributions, respectively.

To protect user privacy and comply with legal requirements, the data provider aggregated individual mobility records into spatial grids with a spatial resolution of 1 km x 1 km and a temporal resolution of one hour. Each grid data record contains attributes, including Grid ID, Grid Location, Timestamp, and Number of Users.

In this study, the term "population count" refers to the hourly number of mobile phone signals detected within each spatial grid, reflecting the static spatial distribution of the population at different times. However, the dataset does not contain individual trajectories or explicit movement paths, and thus differs fundamentally from conventional origin-destination (OD) datasets. Accordingly, this study explores how population movement patterns in disaster contexts can be inferred from such aggregated data, despite the absence of explicit trajectory information.

**Data processing.** Ideally, each grid in the dataset should contain complete hourly data for two full days—August 2 and August 9, 2017—resulting in two 24-hour time series. However, due to unknown reasons, most grids experienced varying degrees of missing values in their time series. To ensure data reliability and accurately capture population dynamics associated with the disaster, the following preprocessing steps were implemented:

(1)  Exclusion of grids with excessive missing data

Grids with three or more consecutive missing hourly values were removed. In the context of an earthquake, human behavior can change drastically within a span of three consecutive hours (12.5% of the total time series), potentially leading to significant shifts in population numbers within a grid. To ensure data reliability, we removed grids with three or more consecutive missing hours. After this filtering step, we retained grid cells with data for both observation dates: August 2, 2017 (six days before the earthquake) and August 9, 2017 (the day after the earthquake). This ensured consistency in the spatial scope of the analysis and improved the accuracy of assessing the earthquake's impact on population. Based on this filtered dataset, we calculated the difference in population counts between the two dates for each grid to generate population anomaly data for further analysis.

(2)  Interpolation of remaining missing values

For grids retained after filtering, linear interpolation was applied to fill isolated missing values. This method was selected for its simplicity and minimal assumptions regarding data behavior, making it suitable for unpredictable conditions such as disaster events. Since grids with excessive missing data had already been removed, the interpolation process was applied under appropriate conditions and yielded reliable results.

(3)   Derivation of population anomalies

To isolate population changes attributable to the earthquake, it was necessary to remove daily variability from the post-disaster data. This was achieved by subtracting the pre-disaster population values (August 2) from the corresponding post-disaster values (August 9) for each hour in each grid. The population anomaly time series were calculated using the following formula:

$$D_{it} = P_{post,it} - P_{pre,it}$$

(1)

where $P_{pre,it}$ and $P_{post,it}$ denote the population counts in grid i at time t on August 2 and August 9, respectively, and $D_{it}$ denotes the population anomaly.

Prior to data processing, we identified 683 grid cells with mobile location data available for at least one timestamp. Among these, approximately 46% were classified as valid grids after the above data procession steps (i.e., with less than three consecutive missing values in their time series), while the remaining 54% were considered invalid. Although invalid grids were more numerous, they accounted for only 13.8% of the total population on August 2. Since both August 2 and

 

August 9 were Wednesdays, population patterns are expected to be similar. Given the low pre-earthquake population in invalid grids, it is reasonable to assume that their post-earthquake population was also limited. Moreover, Fig 2 shows that valid and invalid grids exhibit similar geographic distribution patterns. These findings support the conclusion that excluding invalid grids is unlikely to introduce significant spatial or demographic bias.

Subsequently, we overlaid the spatial distribution of grids onto the official macroseismic intensity map published by the China Earthquake Administration to visualize the seismic intensity level of each grid (Fig 2). This intensity map follows the Chinese seismic intensity standard, under which regions with an intensity level below VI are generally not considered to suffer destructive seismic impacts and are typically excluded from seismic fortification planning [27,28]. Based on this standard, our study focuses on areas where the seismic intensity is VI or above, where the population is more likely to be exposed to potential earthquake impacts. These high-intensity zones are centered around Jiuzhaigou County, with a spatial coverage defined by a latitude range of 32.6128° to 33.6702° and a longitude range of 103.0779° to 104.3304°, as depicted in Fig 2.

## Method

**Empirical orthogonal function (EOF).** EOF analysis is a widely used technique for extracting primary patterns from complex datasets and is particularly effective in analyzing spatiotemporal data. This method decomposes the original dataset into orthogonal spatial and temporal modes by calculating the eigenvalues and corresponding eigenvectors of the covariance matrix derived from the data [29]. The eigenvectors represent the spatial modes (EOFs), which capture the spatial distribution characteristics of the dataset, while the associated eigenvalues indicate the proportion of variance explained by each mode. Higher variance implies greater significance of that spatial mode [30]. The temporal modes, or principal components (PCs) are projections of the original data onto the EOFs, reflecting the temporal evolution of each spatial mode.

To perform EOF analysis, we first constructed a spatiotemporal matrix $O(G, T)$ using the anomaly dataset, where G denotes spatial grids and T denotes hourly timestamps on August 9, 2017. This matrix was mean-centered to produce the anomaly matrix $O'(G, T)$, from which we computed the corresponding covariance matrix $C(G, G)$. Eigenvalues and eigenvectors were then extracted from this matrix to identify the leading EOF modes that account for the largest proportion of variance in the dataset.

Interpretation of EOF results involves examining both the spatial and temporal components. In the spatial modes, positive and negative values indicate grids where the variable (i.e., population count) deviates from the mean in opposite directions, indicating spatial patterns of population inflow or outflow. While the signs of the EOFs and PCs are mathematically arbitrary—multiplying both by −1 does not alter the result—their physical interpretation depends on their combination. For example, a positive PC value associated with a positive EOF anomaly suggests population increase in those grids, whereas a negative EOF anomaly would suggest a decrease. The interpretation reverses when the PC is negative.

In practice, the first few EOF modes typically explain the majority of variance in the dataset and are thus the most informative for understanding dominant spatiotemporal patterns. In our case, the first and second EOF modes alone explained a substantial portion of total variance and revealed meaningful spatial-temporal behaviors of population movement following the 2017 Jiuzhaigou earthquake. Consequently, we focused on the leading EOF modes to extract and interpret the dominant trends of post-disaster population dynamics. The main procedures of the EOF computation are summarized in Table 1. For more details about EOF analysis, please refer to the works of Monahan [30] and Wilks [31].

## The Bregman block average co-clustering algorithm with I-divergence (BBAC_I)

The Bregman Block Average Co-clustering algorithm with I-divergence (BBAC_I) is a spatiotemporal co-clustering technique that simultaneously partitions both rows and columns of a data matrix by minimizing a generalized form of Bregman divergence—specifically, the I-divergence—between the original data and its co-clustered representation. Unlike

**Table 1. Execution Steps of the EOF Algorithm.**

**Empirical orthogonal function**

Input:$O(G, T)$
Start:
1. Subtract the row mean $\mu$ from each row of $O(G, T)$ to generate the anomaly matrix $\acute{O}(G, T)$:
$\acute{O} = \acute{O} - \mu$
2. Generate the covariance matrix $C(G, G)$ based on $\acute{O}(G, T)$:
$C(G, G) = \acute{O}^T * \acute{O}$
3. Decompose $C(G, G)$ to get the eigenvalues $V$ and corresponding eigenvectors $P$:
$C = PVP^{-1}$
4. Sort the eigenvalues in descending order to get $V_{sorted}$ and select the first corresponding eigenvectors as EOFs:
$EOFs = V_{sorted[:, :k]}$
5. Calculate the temporal modes PCs:
$PCs = \acute{O} * EOFs$
End: Output EOFs and PCs

traditional clustering methods that focus solely on spatial similarities and often neglect temporal sequence structures [32], BBAC_I integrates information from both spatial and temporal dimensions, thereby capturing complex patterns across both axes. This method is even advantageous in scenarios involving only spatial clustering, as it incorporates temporal clustering information into spatial clustering, and vice versa [33]. Previous studies [34,35] have validated BBAC_I's effectiveness in revealing key spatial and temporal dynamics in various datasets.

In our analysis, we applied BBAC_I to the forementioned spatiotemporal matrix $O(G, T)$. To focus on spatial pattern differentiation, all time points were grouped into a single temporal cluster, allowing for a uniform comparison of spatial grids across the entire time series. The algorithm then performed co-clustering on the rows and columns of the matrix to produce an initial co-cluster matrix $\hat{O}(\hat{G}, T)$.

The optimization objective of BBAC_I is to minimize the information divergence between the original and co-clustered matrices. The corresponding loss function is defined as the following formula:

$$F_{loss} = D_I (O(G, T) \| \hat{O}(\hat{G}, T))$$

(2)

where $D_I$ denotes the I-divergence, a form of generalized Bregman divergence. Through iterative updates of cluster assignments, the algorithm seeks to minimize $F_{loss}$, thereby enhancing intra-cluster similarity and inter-cluster differentiation. This optimization ensures that the average values within each cluster are internally consistent yet distinct from others. The implementation steps of the BBAC_I algorithm used in this study are summarized in Table 2, for more details on the algorithm, please refer to the studies by Wu [35], Dai [26] and Nattino [36].

## Result

### Population changes under different intensity levels

To assess the earthquake's impact on population dynamics across different seismic intensity zones, we analyzed changes in population anomalies, which represent deviations from typical population fluctuations specifically attributed to the earthquake event. This analysis helped identify key evacuation moments and the overall evacuation process. Fig 3 illustrates the hourly average population anomalies on August 9 (the day following the earthquake), highlighting significant disruptions caused by the earthquake across various intensity levels.

Notably, at 00:00 on August 9 (three hours after the earthquake), the average population anomalies across different intensity zones were 50–200 people higher than usual. This initially elevated population in the most severely affected

**Table 2. Pseudocode of the BBAC_I Algorithm.**

| The Bregman block average co-clustering algorithm with I-divergence |
| --- |

Input:

 $O(G, T)$: Original spatiotemporal matrix with G spatial grids and T timestamps

 tc: Number of time clusters

 gc: Number of grid clusters

 max_iterations: Maximum number of iterations

Procedure:

1 Initialization

 1.1 Perform min_max scaling on $O(G, T)$

 *1.2 Randomly assign grids to gc grid clusters// create initial* $\hat{O}(\hat{G}, T)$

 1.3 tc = 1// All time points are grouped into a single cluster

2 Repeat until f_loss < threshold or iteration count reaches max_iterations

 2.1 Calculate I-divergence:$f_{loss} = D_I(O(G, T) \parallel \hat{O}(\hat{G}, T))$

 2.2 For each grid in G:

  2.2.1 Temporarily assign the grid to each cluster in (1, …, gc)

  2.2.2 Calculate I-divergence for each assignment

  2.2.3 Assign the grid to the cluster that minimizes the I-divergence:

$$i = arg\ min_{i \in \{1,...,gc\}}(floss)$$

 2.3 Update $\hat{O}(\hat{G}, T)$ based on new assignments

Output:

 $\hat{O}(\hat{G}, T)$// Final optimal co-clustered matrix

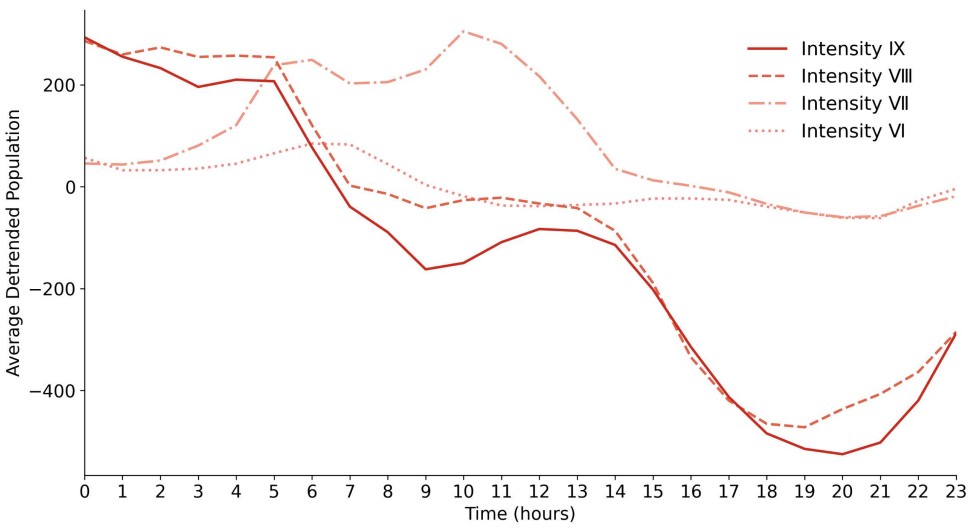

**Fig 3. Hourly average population anomaly across different seismic intensity levels on August 9 (Post-Earthquake).** "Population anomaly" represents deviations from the baseline on August 2 (Pre-Earthquake), highlighting changes specifically attributed to the earthquake, where positive values indicate increases above the baseline and negative values indicate decreases below it.

areas may reflect temporary congregation or delayed evacuation behaviors. Despite this early influx, a noticeable decline in the population anomaly occurred from 00:00–06:00 in the highest intensity zones (Intensity VIII and IX), with a sharper decrease after 06:00. Although some fluctuations were observed, the overall trend remained downward. By 19:00–20:00, the population anomalies in these zones had further decreased by an average of 500–600 compared to normal levels. After 20:00, the population gaps relative to normal days began to narrow, possibly due to the naturally lower nighttime population in the region, which reduced deviations from baseline levels during this period..

Interestingly, the population anomaly in Intensity VII showed an inverse pattern compared to Intensity VIII and IX. Between 00:00 and 06:00, as the population anomaly decreased in Intensity VIII and IX, Intensity VII experienced a delayed increase, indicating a time-lagged population shift from higher to lower intensity zones. After 14:00, the population in Intensity VII showed minimal differences from regular days, suggesting that most evacuees had left the highest-intensity areas.

Intensity VI followed a similar pattern to Intensity VII, though less pronounced. From 00:00–06:00, the average population anomaly increased slightly, by about 50, suggesting a gradual relocation from high-intensity zones to Intensity VI. After 06:00, the population anomaly decreased and remained stable, likely due to the larger grid area of Intensity VI, which covers more than twice the combined grid area of Intensity VII, VIII, and IX, thereby diluting the effect of population inflow.

Overall, these trends indicate that the main period for population gathering and evacuation from the highest intensity zones (Intensity VIII and IX) occurred between 00:00 and 13:00, followed by delayed population increases in adjacent lower-intensity (Intensity VI and VII) areas, indicating a time-lagged relocation process. As the evacuation continued, population in Intensity VI and VII also began to decline. By 14:00, anomalies in these lower-intensity zones had largely returned to normal, suggesting that the majority of the population had evacuated from the highest-intensity areas.

These observed patterns align with the official rescue report [37], which documented several hotels served as temporary shelters in Intensity VIII and IX zones and early rescue operations in the Jiuzhaigou Scenic Area starting at midnight on August 9. This supports the evacuation timeline inferred from our dataset, including the early population decline in the highest-intensity zones and corresponding delayed increases in adjacent lower-intensity areas (Intensity VII and VI).

While this analysis outlines key evacuation trends across seismic intensity zones, it remains challenging to determine the specific evacuation directions, hotspots and evacuation intensity at finer spatial resolutions. Therefore, further analysis at the grid scale is necessary to better understand population movement during the disaster.

## EOF analysis results

Significant disparities in population distribution and evacuation behavior can be observed among grids experiencing the same seismic intensity. Therefore, a grid-level analysis was conducted to elucidate population evacuation dynamics following the earthquake. Consistent with previous sections, we applied EOF analysis to post-earthquake anomaly data from August 9. The first and second modes (EOF1 and EOF2) accounted for 60.4% and 28.5% of the total variance, respectively. To clearly describe the results, we focused on the top 20% of grids with the strongest anomalies in either EOF1 or EOF2, as they accounted for over 80% of the variance in their respective modes.

### (a) EOF1

EOF1 (Fig 4A) reveals a pattern dominated by positive anomalies, with negligible negative anomalies limited to only three grids in Zhangzha and Shuanghe. The strongest positive anomalies were observed in Zhangzha, centered around the epicenter, followed by Chuanzhushi and Shuanghe. The clustering of strong positive anomalies in the Jiuzhaigou Valley demonstrates that EOF1 reflects the most significant population shifts that occurred in this area.

The primary component of EOF1 (PC1), as depicted in Fig 4C, reflects the dynamic population trends within the Jiuzhaigou Valley. Between midnight and 5 AM, PC1 showed a consistent increase, indicating a continuous inflow of people—likely seeking temporary refuge within the valley. After 5 AM, PC1 began to decline but remained positive, suggesting a decelerated rate of inflow, potentially reflecting the initial phase of evacuation, that some individuals may had already started evacuating. Following 12 PM, PC1 continued to decline and turned negative, reflecting a substantial population outflow, consistent with the official report of large-scale evacuations from Jiuzhaigou County during this period [37]. This correspondence confirms EOF1 as a reliable indicator of major population movements following the earthquake.

### (b) EOF2

EOF2 captured the spatial heterogeneity of population shifts during the evacuation process. The spatial pattern (Fig 4B) exhibited a pronounced contrast between grids with positive and negative anomalies. Most grids near the People's

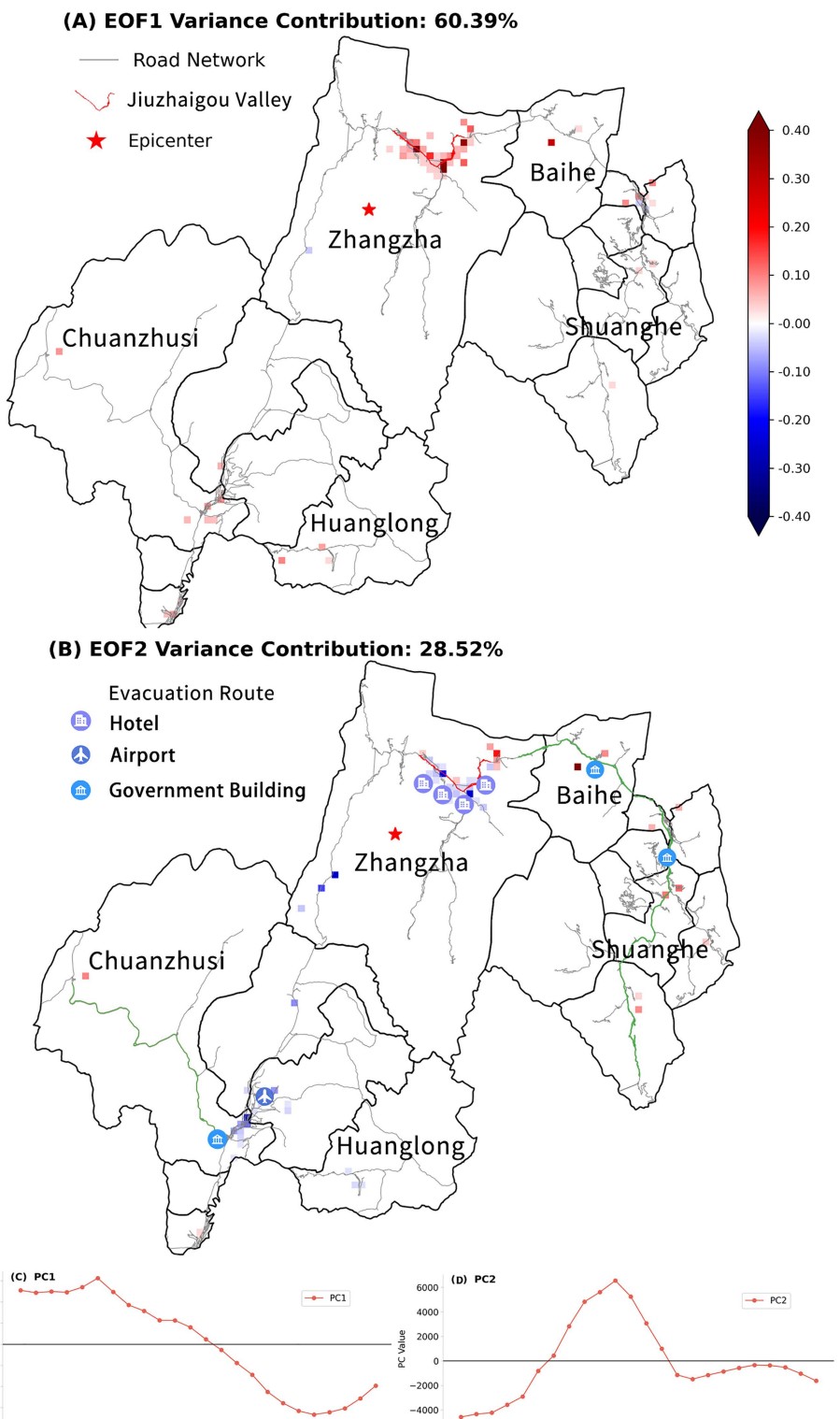

**Fig 4. The EOF analysis results on August 9.** (A-B) Spatial distribution of EOF1 and EOF2, respectively. (C-D) PCs curves corresponding to EOF1 and EOF2, respectively.

Government of Chuanzhushi (marked as a government building icon in Fig 4B) were featured with negative anomalies, except for one positive anomaly located in the northwest of Chuanzhushi. Another core area of negative anomalies was located in the Jiuzhaigou Valley (Zhangzha Town). Extending eastward from the valley and then southward toward Shuanghe Town, the surrounding grids exhibited a continuous spatial corridor formed by positive anomalies.

The primary component of EOF2 (PC2), as depicted in Fig 4D, offers insight into the evolving evacuation dynamics. From 0 to 6 AM, PC2 gradually increased but remained negative, indicating a gradual slowdown in population accumulation in negatively anomalous areas such as grids near the People's Government of Chuanzhushi and the Jiuzhaigou Valley. After 6 AM, PC2 turned positive and rose sharply, suggesting a significant shift as individuals evacuated toward positively anomalous regions. The peak at 10 AM represents the height of evacuation activity, followed by a steady decline, with PC2 approaching zero around 2 PM—suggesting that most evacuations had been completed by then.

Based on the temporal evolution of PC2 and the spatial distribution of EOF2 anomalies, we infer that evacuation routes during the earthquake corresponded to the pathways connecting negative anomaly centers (evacuation origins) to positive anomaly zones (potential shelters or transit hubs). We then overlaid the EOF2 anomaly map with the local road network (Fig 4B) and identified two primary evacuation routes: Route 1 connected the negative anomaly grids near the Chuanzhusi government building to the isolated positive anomaly grid in the northwest (green line on the left of Fig 4B); Route 2 extended from the negative anomaly center in the Jiuzhaigou Valley toward the positively anomalous region in Shuanghe to the south (green line on the right of Fig 4B). Both routes align closely match those documented in the official rescue report [37], with Route 2 being confirmed as the principal evacuation corridor.

To further validate the inferred spatial patterns, we mapped real-world points of interest (POIs) identified from evacuation reports and media sources such as People's Daily, Sichuan Daily onto the EOF2 anomaly map (Fig 4B). Major evacuation hubs—such as Chuanzhusi Government Square, Jiuzhaigou Huanglong Airport and hotels distributed throughout Jiuzhaigou Valley—were located within areas of strong negative anomalies and served as initial gathering sites. Conversely, sites near zones of strong positive anomalies, such as the Baihe County and Jiuzhaigou County government buildings, functioned as transit or secondary shelter sites. These correspondences reinforce the robustness of the EOF2-derived evacuation interpretations.

Together, EOF1 and EOF2 provide a comprehensive understanding of population dynamics during the earthquake. While EOF1 effectively identifies regions experiencing the most significant population shifts, EOF2 captures the spatio-temporal characteristics of evacuation behavior, offering detailed insights into the evacuation process.

## Comparison of EOF and BBAC_I in analyzing population change

Previous research has employed the BBAC_I clustering method to analyze population changes during the Jiuzhaigou earthquake [26]. To enable a direct comparison with the Empirical Orthogonal Function (EOF) method, we applied the BBAC_I method to the same dataset.

Since the BBAC_I method cannot handle negative data, we used the original population data for clustering analysis. Fig 5 provides a detailed spatial distribution of each cluster, along with corresponding time series of population counts on August 9 (Post-Earthquake). Overall, the cluster distribution shows a radial pattern centered on Cluster 3 (highlighted in red), with the magnitude of population change diminishing outward.

Cluster 3, serving as the core of the overall spatial distribution, contained the highest population in the fewest grids. In these grids, the population remained stable until 5 AM, followed by a decline between 5 and 10 AM, and then a more gradual decrease thereafter. An exceptional pattern was observed in one grid within Cluster 3, where the population surged to approximately 8,000 between midnight and 10 AM before rapidly declining, with the population approaching zero by 2 PM—indicating near-complete evacuation.

Cluster 2 and Cluster 4, surrounding Cluster 3, comprise the majority of the spatial grids. Most grids in Cluster 4 experienced a population decline between 5 and 10 AM, followed by more moderate changes later in the day. Some grids,

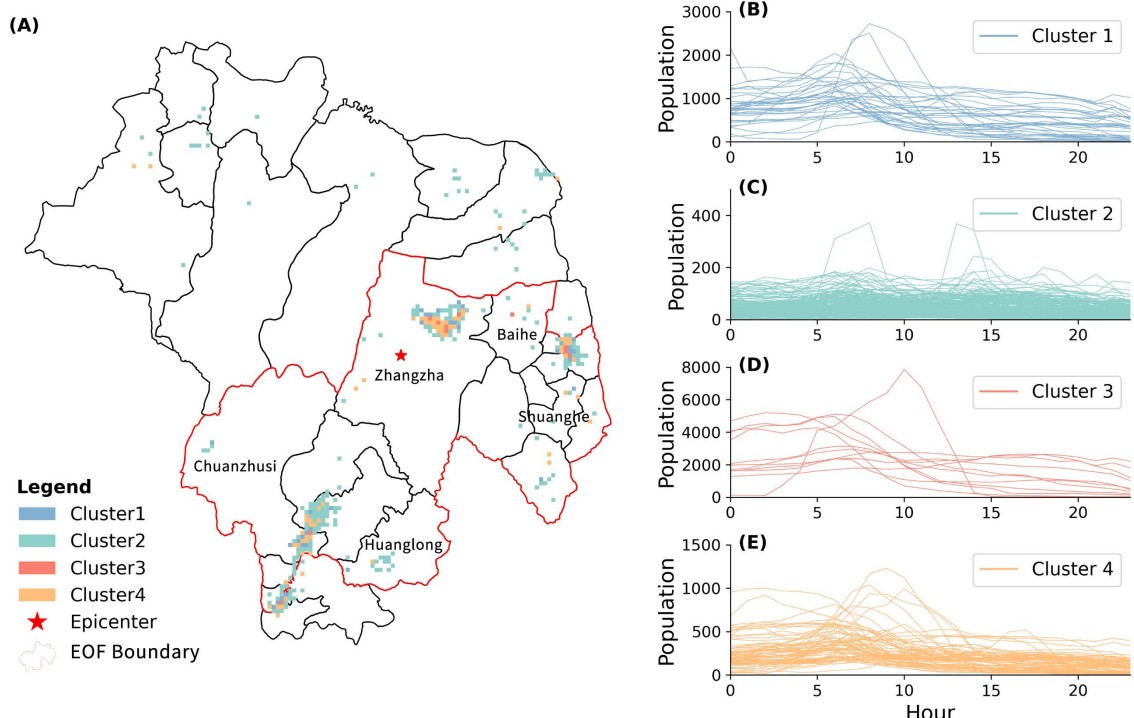

**Fig 5. Cluster analysis results on August 9. (A)** Spatial distribution of clusters. **(B-E)** Temporal population series for grids corresponding to cluster1-4, respectively.

however, showed a population increase during this same interval, with a few continuing to grow afterward. Cluster 2, typically located on the peripheries of towns, exhibited smaller populations and weaker fluctuations, with no grid exceeding 400 individuals during the analyzed period.

Cluster 1, characterized by fewer and more randomly distributed grids. These grids showed a noticeable population increase in the early morning hours and peaked between 5 AM and 10 AM, followed by a rapid decline.

While the BBAC_I method effectively integrates temporal patterns into spatial clustering and shows proficiency in deriving clusters with similar temporal patterns, it has limitations. Specifically, it does not clearly identify the most severely impacted areas or support preliminary assessment of evacuation directions in seismic events.

In contrast, the EOF method effectively overcomes these limitations. By focusing on the top 20% of the strongest anomalies in EOF1 and EOF2, we successfully identified regions with the most significant changes in population activity. For example, grids in Cluster 2 were largely absent from the EOF results due to minimal variation, thus helping to pinpoint the key areas that require prioritized emergency attention. In addition, PC2 captured both the timing and magnitude of population movements among grids, providing an initial classification of grid functions during the earthquake, such as evacuation corridors and temporary shelter areas.

Furthermore, EOF analysis aids in identifying evacuation routes even without explicit tracking data. For example, although Cluster 3 exhibited the largest and most dramatic population changes, the direction and extent of evacuation from these areas remained unclear. EOF results revealed that a significant number of individuals evacuated from Cluster 3 in the Jiuzhaigou valley and passed through Shuanghe Town, thereby suggesting a probable evacuation corridor. This finding underscores the value of the EOF method in identifying evacuation routes and priority zones for emergency response.

In summary, compared to the BBAC_I method, the EOF method provides a more comprehensive understanding of population dynamics during disaster events. By reducing multidimensional data into a few principal modes, it effectively highlights both key temporal patterns in population activity and major spatial movements related to population evacuation.

## Discussion

### Impact of missing data on the study

In this study, we noticed an unusual phenomenon: shortly after the earthquake, most grid areas, including those in high-intensity seismic zones and near the epicenter, exhibited population increases, as detailed in Fig 3 and Fig 5. This observation is counterintuitive, as people are generally expected to evacuate high-risk areas following a disaster. Prompted by this anomaly, we conducted a thorough reassessment of our data preprocessing procedures.

During data preprocessing, we excluded grid data with a large number of missing values in their time series. Our analysis indicates that such data loss was not random but likely associated with severe earthquake-induced disruptions—particularly the destruction of base stations and power outages, which are the most plausible causes of the missing population data in the time series. Consequently, although located in the same intensity zones as their neighboring grids with valid data, these excluded grids were probably the most heavily impacted [14,38].

The presence of stable signal coverage likely reflected the continued functionality of critical infrastructure—such as communication and power networks—and may have influenced survivors' evacuation decisions. As a result, populations may have relocated from areas with disrupted signal coverage (excluded from analysis) to nearby areas with available signal coverage (included in analysis), which were perceived as relatively safer. This dynamic could explain the counterintuitive population inflow observed within the highest-intensity zones, as survivors concentrated in grids with better infrastructure conditions.

Therefore, our findings suggest that in practical rescue operations, special attention should be paid to areas with extensive missing mobile location data. The absence of such data could be an indicator of more severe disaster impacts in these locations.

## Conclusion

Our study demonstrates the utility of grid-aggregated mobile location data with advanced analytical techniques—particularly the Empirical Orthogonal Function (EOF) method—to analyze population dynamics during disasters. Through the EOF method, we were able to preliminarily identify the actual evacuation routes following the Jiuzhaigou earthquake, revealing patterns of population movement. Although the data do not provide individual trajectory information, they still offer meaningful insights into large-scale evacuation behaviors.

The main findings derived from our analysis are summarized as follows:

(1) EOF1 captured the overall population evacuation patterns. It revealed that the regions with strong population changes after the earthquake, such as Zhangzha, Chuanzhushi, and Shuanghe, experienced the most significant evacuations.

(2) EOF2 provided more detailed insights into the specific evacuation routes and variations in population movement. It identified critical evacuation paths in heavily affected areas like Zhangzha and Chuanzhushi, highlighting how populations evacuated from higher-risk areas. The analysis of PC2 revealed shifts in evacuation intensity over time, offering a clearer picture of the evacuation flow dynamics.

(3) Compared to the BBAC_I method, the EOF approach offers a deeper understanding of disaster-induced population dynamics by filtering out low-impact grids, identifying functional zones such as evacuation corridors and shelters, and capturing both the temporal and spatial characteristics of evacuation without relying on individual trajectory data.

While our study provides valuable insights, there are several areas for optimization in future research. First, the analysis was limited by the availability of only two observation days in the dataset: August 2, 2017 (six days before the earthquake), and August 9, 2017 (the day after the earthquake). This restricted temporal coverage may introduce inaccuracies when removing daily variation, reducing the overall validity of the results to some extent. Second, missing mobile location data could be further utilized to assess damage to power systems during disasters and pinpoint the most hazardous areas. Additionally, the estimation of evacuation routes could be enhanced by employing advanced techniques, such as complex network analysis, to further uncover the actual paths taken by evacuees. This approach would provide more detailed information on the direction and intensity of population movements across different areas, leading to more accurate and reliable evacuation route estimations.

## Acknowledgments

The authors gratefully acknowledge the support of Dr. Shi Shen from Beijing Normal University for providing the mobile phone location data used in this study.

## Author contributions

**Conceptualization:** Zezhi Lin.

**Data curation:** Saini Yang, Po Pan.

**Formal analysis:** Zezhi Lin.

**Funding acquisition:** Rui Mao.

**Investigation:** Zezhi Lin.

**Methodology:** Zezhi Lin, Rui Mao.

**Project administration:** Rui Mao.

**Software:** Huaiqun Zhao.

**Supervision:** Rui Mao.

**Validation:** Zezhi Lin, Rui Mao.

**Visualization:** Zezhi Lin, Huaiqun Zhao, Zihui Tang.

**Writing – original draft:** Zezhi Lin.

**Writing – review & editing:** Rui Mao.

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
