## [Decision Letter · Decision Letter 0]

17 May 2025

Dear Dr. Mao,

We look forward to receiving your revised manuscript.

Kind regards,

James Colborn

Academic Editor

PLOS ONE

Journal Requirements:

This research was supported by the National Key Research and Development Program (2022YFC3004404).

4. In the online submission form, you indicated that the data underlying the results presented in the study are available from Shishen at Beijing Normal University (Contact E-mail:shens@bnu.edu.cn).

5. Please amend your list of authors on the manuscript to ensure that each author is linked to an affiliation. Authors’ affiliations should reflect the institution where the work was done (if authors moved subsequently, you can also list the new affiliation stating “current affiliation:….” as necessary).’

6. We note that Figure 1, 3, 4, 5, in your submission contain [map/satellite] images which may be copyrighted. All PLOS content is published under the Creative Commons Attribution License (CC BY 4.0), which means that the manuscript, images, and Supporting Information files will be freely available online, and any third party is permitted to access, download, copy, distribute, and use these materials in any way, even commercially, with proper attribution. For these reasons, we cannot publish previously copyrighted maps or satellite images created using proprietary data, such as Google software (Google Maps, Street View, and Earth). For more information, see our copyright guidelines: http://journals.plos.org/plosone/s/licenses-and-copyright.

a. You may seek permission from the original copyright holder of Figure 1, 3, 4, 5, to publish the content specifically under the CC BY 4.0 license.

Reviewers' comments:

Reviewer's Responses to Questions

**Comments to the Author**

1. Is the manuscript technically sound, and do the data support the conclusions?

Reviewer #1: Yes

Reviewer #2: Partly

2. Has the statistical analysis been performed appropriately and rigorously?

Reviewer #1: Yes

Reviewer #2: I Don't Know

3. Have the authors made all data underlying the findings in their manuscript fully available?

Reviewer #1: Yes

Reviewer #2: Yes

4. Is the manuscript presented in an intelligible fashion and written in standard English?

Reviewer #1: No

Reviewer #2: Yes

Reviewer #1: The manuscript employs the Empirical Orthogonal Function (EOF) method on mobile phone location data to analyze population movements following the 2017 Jiuzhaigou earthquake. It compares these results with those obtained using the BBAC_I clustering method. The topic of the paper has practical significance for post-disaster emergency rescue, and evacuation route planning.

Overall, the manuscript demonstrates a higher levels of innovation and practical value. However, the paper could be further improved by addressing some limitations and gaps in the analysis and discussion. The following suggestions are offered:

1. It is recommended to add an analytical framework figure and a location map of the study area, to clarify the spatial context of the analysis.

2. In the introduction, the fourth paragraph presents the advantages of the EOF method and states its use in this study, while the beginning of the fifth paragraph points out the shortcomings of the current method, BBAC_I. It is suggested that the authors first discuss the deficiencies of the currently adopted method, and then summarize the advantages of the EOF method and explain why it was chosen.

3. In Section 2.2.1(1), regarding 'Eliminate grids with three or more consecutive missing values in their time series', the grids that did not capture data did so for unknown reasons. Could these unknown reasons affect the study’s results? It is recommended to include a spatial distribution map of the data to more clearly demonstrate its distribution. And visualize the spatial distribution of missing grids to assess whether their exclusion introduces regional bias.

4. The manuscript states, 'After removing these grids, we performed an inner join on the remaining grids to ensure that only spatial regions with comparable data before and after the disaster were included in the analysis.' What exactly is meant by 'inner join' in this context?

5. In Fig 1, 'Spatial intensity distribution on August 9, 2017', is the intensity describing the grid cells? It is recommended to add the official intensity map for comparison. Moreover, the figure does not clearly show the comparison between the anomaly data and the macroseismic intensity.

Currently, the figure’s caption is ambiguous. It is unclear whether the colors represent the actual seismic intensity levels or the magnitudes of the data-derived anomalies.

6. It is suggested to merge the infrastructure POIs shown in Fig 5 into Fig 3(a) and (b) to reduce the number of figures.

7. “In particular, regions with an earthquake intensity below VI generally do not suffer destructive seismic impacts and are not typically considered in seismic fortification (Nazmfar et al. 2019; Li et al. 2021). Therefore, our study area encompasses regions where the seismic intensity is above V, representing areas at greater risk of damage”. Does the seismic intensity above V represents areas at greater risk of damage?

8. Language and writing need to be further improved.

9. The format of the reference citation in the main text is not standard, please revise them.

Reviewer #2: I appreciate the opportunity to review this manuscript, “Analysis of Disaster-Affected Population Mobility Through Grid-Aggregated Mobile Location Data: The 2017 Jiuzhaigou Earthquake, China.” This is a really strong paper with interesting insights on methods to understand population movement after a disaster.

A few overarching comments:

1) The abstract is very clear. However, it does not mention the BBAC_I method which is a substantial part of your results and methods and the conclusion, therefore, does not compare the results across the EOF1, EOF2, and BBAC_I methods. This reviewer recommends enhancing the abstract to reflect all the methods used and the more robust conclusion you reached.

2) When did you access the mobile phone data? Or, more directly, when was the data you accessed available? I appreciate your emphasis in the introduction that the analyses you completed could be implemented quickly after a disaster. However, I do not see a detailed description of the opportunity to complete these analyses soon after the earthquake. Please provide more detail about when the data is available and when you completed the analyses.

3) There are a few results and discussion topics that are not introduced in the methods section. For example, the positive and negative EOF analyses, the Intensity zones, and the BBAC_I clusters are not presented in the methods. Please ensure that you describe in the methods all the ways that you reach the results.

4) Lines 233 – 240, 270 – 274, and 434 - 440 describe interpretations of data that lead beyond the scope of the described study. Please consider removing descriptions of data interpretations that do not directly follow from the analytic methods your study implemented.

5) The discussion presents some results that were not described in the results section. Please ensure all results are mentioned in the results section.

Specific suggestions

1) Lines 94 – 96 describe the structure of a standard scientific paper. Therefore, please consider removing these lines.

2) Lines 109 and 151, at least, references using data from August 2, the day before the earthquake. If the earthquake was on August 8, how is August 2 the day before the earthquake? Please review this detail.

3) Can you please increase the amount of detail you provide about missingness? I understand that some grids have missingness and that you removed those, but I am not fully clear on how much missingness there was. Please share more about what proportion of grids you had to remove.

4) Please consider removing lines 217 and 218 as the sentence describes the conventional purpose of a results section.

5) Line 268 references “the report”. Please ensure that the reader knows exactly what ‘report’ you are referencing.

Thank you very much for this opportunity to review a very interesting paper.

**Do you want your identity to be public for this peer review?** For information about this choice, including consent withdrawal, please see our Privacy Policy

Reviewer #1: No

Reviewer #2: No

---

## [Author Response · Author response to Decision Letter 1]

17 Jul 2025

Response to Reviewers

We would like to express our sincere gratitude to the editor and the two reviewers for their constructive feedback on our manuscript. We have carefully considered each of their comments and made substantial revisions throughout the manuscript, including refinements to the figures and language to improve clarity and eliminate potential ambiguities. In addition, the analysis has been further refined where appropriate. All changes are marked in blue in the revised manuscript, which you can see in the additional material. Below are our item-by-item responses.

Response to Editor

Comment 1: Please ensure that your manuscript meets PLOS ONE's style requirements, including those for file naming.

Response 1: Done. We have revised the manuscript to fully comply with PLOS ONE's style requirements.

Comment 2: Thank you for stating the following financial disclosure:

This research was supported by the National Key Research and Development Program (2022YFC3004404).

Response 2: Thank you for your guidance. We have added the required statement— “The funders had no role in study design, data collection and analysis, decision to publish, or preparation of the manuscript.”—to our cover letter.

Comment 3: We note that you have indicated that there are restrictions to data sharing for this study. For studies involving human research participant data or other sensitive data, we encourage authors to share de-identified or anonymized data. However, when data cannot be publicly shared for ethical reasons, we allow authors to make their data sets available upon request. If there are ethical or legal restrictions on sharing a de-identified data set, please explain them in detail (e.g., data contain potentially identifying or sensitive patient information, data are owned by a third-party organization, etc.) and who has imposed them (e.g., a Research Ethics Committee or Institutional Review Board, etc.). Please also provide contact information for a data access committee, ethics committee, or other institutional body to which data requests may be sent.

Response 3: Thank you for your comments. The mobile phone location data used in this study were obtained in 2023 via private communication with Dr. Shen at Beijing Normal University. We are not authorized to share the dataset publicly, as the data are held by Dr. Shen

Comment 4: In the online submission form, you indicated that the data underlying the results presented in the study are available from Shishen at Beijing Normal University (Contact E-mail: shens@bnu.edu.cn). This policy applies to all data except where public deposition would breach compliance with the protocol approved by your research ethics board. If your data cannot be made publicly available for ethical or legal reasons (e.g., public availability would compromise patient privacy), please explain your reasons on resubmission and your exemption request will be escalated for approval.

Response 4: Thank you for your comments. After discussion with Dr. Shi Shen, we confirm that the mobile phone location data used in this study are subject to legal restrictions and cannot be publicly shared.

The dataset was originally acquired from China Unicom under a data use agreement that includes strict confidentiality clauses. According to the terms of this agreement, Dr. Shen is prohibited from sharing the data without formal authorization from China Unicom. Therefore, the authors do not have the legal right to make the data publicly available.

Interested researchers may request access to the dataset by contacting:

Dr. Shi Shen (Beijing Normal University)

Email: shens@bnu.edu.cn

Strategic Customer Department, China Unicom

Phone: +86 18519518331

Access is subject to approval by China Unicom and may require appropriate data use agreements and/or ethical review, depending on the nature of the request.

Comment 5: Please amend your list of authors on the manuscript to ensure that each author is linked to an affiliation. Authors’ affiliations should reflect the institution where the work was done.

Response 5: Done. Please see the details in the revised manuscript on Page 1, Lines 4-10 as follows.

“

Zezhi Lin2, Rui Mao1,*, Huaiqun Zhao1, Zihui Tang3, Saini Yang1, Po Pan4

1 School of National Safety and Emergency Management, Beijing Normal University, Beijing, 100875, China

2 School of Systems Science, Beijing Normal University, Beijing, 100875, China

3 School of Architecture and Urban Planning, Beijing University of Civil Engineering and Architecture, Beijing, 102627, China

4 Unicom (Beijing) Industry Internet Co.,Ltd., Beijing, 100038, China

* Corresponding author

”

Comment 6: We note that Figure 1, 3, 4, 5, in your submission contain [map/satellite] images which may be copyrighted. All PLOS content is published under the Creative Commons Attribution License (CC BY 4.0), which means that the manuscript, images, and Supporting Information files will be freely available online, and any third party is permitted to access, download, copy, distribute, and use these materials in any way, even commercially, with proper attribution. For these reasons, we cannot publish previously copyrighted maps or satellite images created using proprietary data, such as Google software (Google Maps, Street View, and Earth). For more information, see our copyright guidelines: http://journals.plos.org/plosone/s/licenses-and-copyright.

Response 6: Thank you for your comments. The base maps used in Figures 1, 3, 4, and 5 of our manuscript were obtained from the National Platform for Common GeoSpatial Information Services (https://www.tianditu.gov.cn/).

This platform is a government-operated service in China that provides publicly available geospatial data and base maps for academic, educational, and non-commercial purposes. To the best of our knowledge, the use of these maps is permitted under open government data policies, and they are not subject to commercial copyright restrictions like Google Maps or similar proprietary services. Please see the details in the “map_copyright_statement” file, we have uploaded it as an "Other" file with our submission.

Comment 7: Please review your reference list to ensure that it is complete and correct. If you have cited papers that have been retracted, please include the rationale for doing so in the manuscript text, or remove these references and replace them with relevant current references. Any changes to the reference list should be mentioned in the rebuttal letter that accompanies your revised manuscript. If you need to cite a retracted article, indicate the article’s retracted status in the References list and also include a citation and full reference for the retraction notice.

Response 7: Thank you for your comments. All in-text citations and references have been thoroughly checked and corrected to comply with the PLOS ONE formatting guidelines. Please see the details in the revised manuscript on Pages 28-31, Lines 508-616 as follows.

“

1. Butuzova A, Dolgikh V. Study of the target structure of transport mobility of the population on the example of Irkutsk. Transport Technician: Education and Practice. 2023 Sep 9;4:322–31.

2. Opdyke A, Lepropre F, Javernick-Will A, Koschmann M. Inter-organizational resource coordination in post-disaster infrastructure recovery. Construction Management and Economics. 2017 Sep 2;35(8–9):514–30.

3. Hu L, Fang Z, Zhang M, Jiang L, Yue P. Facilitating Typhoon-Triggered Flood Disaster-Ready Information Delivery Using SDI Services Approach—A Case Study in Hainan. Remote Sensing. 2022 Jan;14(8):1832.

4. Liu C, Chen Y, Wei Y, Chen F. Spatial Population Distribution Data Disaggregation Based on SDGSAT-1 Nighttime Light and Land Use Data Using Guilin, China, as an Example. Remote Sensing. 2023 Jun 3;15(11):2926.

5. Li J, He Z, Plaza J, Li S, Chen J, Wu H, et al. Social Media: New Perspectives to Improve Remote Sensing for Emergency Response. Proceedings of the IEEE. 2017 Oct;105(10):1900–12.

6. Yao K, Yang S, Tang J. Rapid assessment of seismic intensity based on Sina Weibo — A case study of the changning earthquake in Sichuan Province, China. Int J Disaster Risk Reduct. 2021 May;58:102217.

7. Kawade SS, Akant K. Real Time Image Processing System for Crop Segmentation. In: 2021 6th International Conference for Convergence in Technology (I2CT); 2021. p. 1–6. doi: 10.1109/I2CT51068.2021.9417921.

8. Lu Z, Long Z, Xia J, An C. A Random Forest Model for Travel Mode Identification Based on Mobile Phone Signaling Data. Sustainability. 2019 Oct 25;11(21):5950.

9. Yabe T, Jones NKW, Rao PSC, Gonzalez MC, Ukkusuri SV. Mobile phone location data for disasters: A review from natural hazards and epidemics. Computers, Environment and Urban Systems. 2022 Jun;94:101777.

10. Yabe T, Tsubouchi K, Sudo A, Sekimoto Y. A framework for evacuation hotspot detection after large scale disasters using location data from smartphones: case study of Kumamoto earthquake. In: Proceedings of the 24th ACM SIGSPATIAL International Conference on Advances in Geographic Information Systems; 2016. p. 44. doi: 10.1145/2996913.2997014.

11. Wilson R, zu Erbach-Schoenberg E, Albert M, Power D, Tudge S, Gonzalez M, Guthrie S, Chamberlain H, Brooks C, Hughes C, Pitonakova L. Rapid and near real-time assessments of population displacement using mobile phone data following disasters: The 2015 Nepal earthquake. PLoS currents. 2016 Feb 24;8:ecurrents-dis.

12. Duan Z, Lei Z, Zhang M, Li W, Fang J, Li J. Understanding evacuation and impact of a metro collision on ridership using large-scale mobile phone data. IET Intelligent Transport Systems. 2017;11(8):511–20.

13. Xia C, Nie G, Fan X, Zhou J, Pang X. Research on the application of mobile phone location signal data in earthquake emergency work: A case study of Jiuzhaigou earthquake. Martínez-Álvarez F, editor. PLOS ONE. 2019 Apr 12;14(4):e0215361.

14. Acosta RJ, Kishore N, Irizarry RA, Buckee CO. Quantifying the dynamics of migration after Hurricane Maria in Puerto Rico. Proc Natl Acad Sci. 2020 Dec 22;117(51):32772–8.

15. Tan S, Lai S, Fang F, Cao Z, Sai B, Song B, et al. Mobility in China, 2020: a tale of four phases. Natl Sci Rev. 2021 Dec 4;8(11):nwab148.

16. Guo X, Wei B, Nie G, Su G. Application of Mobile Signaling Data in Determining the Seismic Influence Field: A Case Study of the 2017 Mw 6.5 Jiuzhaigou Earthquake, China. International Journal of Environmental Research and Public Health. 2022 Jan;19(17):10697.

17. Wei B, Su G, Liu F. Dynamic Assessment of Spatiotemporal Population Distribution Based on Mobile Phone Data: A Case Study in Xining City, China. Int J Disaster Risk Sci. 2023;14:649–665. doi: 10.1007/s13753-023-00480-3.

18. Yabe T, Sekimoto Y, Tsubouchi K, Ikemoto S (2019) Cross-comparative analysis of evacuation behavior after earthquakes using mobile phone data. PLoS ONE 14(2): e0211375. https://doi.org/10.1371/journal.pone.0211375.

19. Hong B, Bonczak BJ, Gupta A, Kontokosta CE. Measuring inequality in community resilience to natural disasters using large-scale mobility data. Nat Commun. 2021 Mar 25;12(1):1870.

20. Deng H, Aldrich DP, Danziger MM, Gao J, Phillips NE, Cornelius SP, et al. High-resolution human mobility data reveal race and wealth disparities in disaster evacuation patterns. Humanit Soc Sci Commun. 2021 Jun 15;8(1):1–8.

21. Rahimi-Golkhandan A, Garvin MJ, Wang Q. Assessing the Impact of Transportation Diversity on Postdisaster Intraurban Mobility. Journal of Management in Engineering. 2021 Jan 1;37(1):04020106.

22. Yabe T, Ukkusuri SV, C. Rao PS. Mobile phone data reveals the importance of pre-disaster inter-city social ties for recovery after Hurricane Maria. Appl Netw Sci. 2019 Dec;4(1):98.

23. Bengtsson L, Lu X, Thorson A, Garfield R, Von Schreeb J. Improved Response to Disasters and Outbreaks by Tracking Population Movements with Mobile Phone Network Data: A Post-Earthquake Geospatial Study in Haiti. Gething PW, editor. Plos Med. 2011 Aug 30;8(8):e1001083.

24. Podesta C, Coleman N, Esmalian A, Yuan F, Mostafavi A. Quantifying community resilience based on fluctuations in visits to points-of-interest derived from digital trace data. Journal of The Royal Society Interface. 2021 Apr;18(177):20210158.

25. Cheng Y, Church G. Biclustering of Expression Data. Proceedings / . International Conference on Intelligent Systems for Molecular Biology ; ISMB International Conference on Intelligent Systems for Molecular Biology. 2000 Feb 1;8:93–103.

26. Dai K, Cheng C, Shen S, Su K, Zheng X, Zhang T. Postearthquake situational awareness based on mobile phone signaling data: An example from the 2017 Jiuzhaigou earthquake. Int J Disaster Risk Reduct. 2022 Feb;69:102736.

27. The Chinese Seismic Intensity Scale. Standards Press of China, Beijing, China; 2020. Available from: https://kns.cnki.net/kcms2/article/abstract?v=-93ivAxQXRqFgn1DcmW2S-9m_QCqsjgrPKcoKZn4pezR3ebCWzCPkcM9WBYLyyY9rhS6aT9nhcK0VBXd2zFTxsi44A_x4zRi41R-iaKEvL8FEVC5Iqo3WZIqIXGwtb8u80S8Bk6zHEpTr6iE58RZQQ==&uniplatform=NZKPT&language=CHS

28. Li S, Chen Y, Yu T. Comparison of macroseismic-intensity scales by considering empirical observations of structural seismic damage. Earthquake Spectra. 2021 Feb 1;37(1):449–85.

29. Hannachi A, Jolliffe IT, Stephenson DB. Empirical orthogonal functions and related techniques in atmospheric science: A review. International Journal of Climatology. 2007;27(9):1119–52.

30. Monahan AH, Fyfe JC, Ambaum MHP, Stephenson DB, North GR. Empirical Orthogonal Functions: The Medium is the Message. Journal of Climate. 2009 Dec 15;22(24):6501–14.

31. Wilks DS. Principal Component (EOF) Analysis. In: Wilks DS, editor. Statistical Methods in the Atmospheric Sciences. 4th ed. Amsterdam: Elsevier; 2019. p. 617–668. doi: 10.1016/B978-0-12-815823-4.00013-4.

32. Wu X, Poorthuis A, Zurita-Milla R, Kraak MJ. An interactive web-based geovisual analytics platform for co-clustering spatio-temporal data. Computers & Geosciences. 2020 Apr 1;137:104420.

33. Banerjee A, Dhillon I, Ghosh J, Merugu S, Modha DS. A generalized maximum entropy approach to bregman co-clustering and matrix approximation. In: Proceedings of the Tenth ACM SIGKDD International Conference on Knowledge Discovery and Data Mining; 2004. p. 509–514. doi: 10.1145/1014052.1014111.

34. Wu X, Zurita-Milla R, Kraak MJ. Co-clustering geo-referenced time series: exploring spatio-temporal patterns in Dutch temperature data. Int J Geogr Inf Sci. 2015 Apr 3;29(4):624–42.

35. Wu X, Cheng C, Qiao C, Song C. Spatio-temporal differentiation of spring phenology in China driven by temperatures and photoperiod from 1979 to 2018. Science China Earth Sciences. 2020 Oct 1;63(10):1485–98.

36. Nattino F, Ku O, Grootes MW, Izquierdo-Verdiguier E, Girgin S, Zurita-Milla R. CGC: a Scalable Python Package for Co- andTri-Clustering of Geodata Cubes. Journal of Open Source Software. 2022 Apr 10;7(72):4032.

37. Chen X. Experience and Lessons of Public Evacuation After the “8·8” JiuZhaigou Earthquake in Sichuan Province. Journal of China Emergency Management Science. 2019;(36):36–43.

38. Zhang X, Lu Q, Ning B, Huang Z. A rapid identification method for severely earthquake-damaged areas based on damaged mobile phone base stations in China. Environ Earth Sci. 2016 Apr 13;75(8):704.

”

Response to Reviewer #1

Comment 1: It is recommended to add an analytical framework figure and a location map of the study area, to clarify the spatial context of the analysis.

Response 1: Thank you for your comments. To enhance clarity, we have added an analytical framework (Fig. 1) and a study area location map (Fig. 2) to illustrate the methodology and spatial setting. Please refer to Figs. 1 and 2 below for details.

Fig. 1 Analytical framework of the study

Fig. 2 Spatial intensity distribution on August 9, 2017

Comment 2: In the introduction, the fourth parag

---

## [Decision Letter · Decision Letter 1]

12 Oct 2025

Analysis of disaster-affected population mobility through grid-aggregated mobile location data: The 2017 Jiuzhaigou Earthquake, China

PONE-D-25-08571R1

Dear Dr. Mao,

We’re pleased to inform you that your manuscript has been judged scientifically suitable for publication and will be formally accepted for publication once it meets all outstanding technical requirements.

Kind regards,

James Colborn

Academic Editor

PLOS ONE

Additional Editor Comments (optional):

Reviewers' comments:

Reviewer's Responses to Questions

**Comments to the Author**

Reviewer #3: (No Response)

Reviewer #4: All comments have been addressed

2. Is the manuscript technically sound, and do the data support the conclusions?

Reviewer #3: Yes

Reviewer #4: Yes

3. Has the statistical analysis been performed appropriately and rigorously?

Reviewer #3: Yes

Reviewer #4: Yes

4. Have the authors made all data underlying the findings in their manuscript fully available?

Reviewer #3: No

Reviewer #4: Yes

5. Is the manuscript presented in an intelligible fashion and written in standard English?

Reviewer #3: No

Reviewer #4: Yes

Reviewer #3: The manuscript presents an interesting and valuable attempt to analyze disaster-affected population mobility during the 2017 Jiuzhaigou Earthquake using grid-aggregated mobile phone location data. The application of the EOF method, and its comparison with BBAC\_I, is innovative and shows potential for addressing the lack of individual trajectory information in aggregated datasets. However, the discussion section is notably weak and underdeveloped. It does not provide a systematic reflection on the strengths and limitations of the study, nor does it adequately outline prospects for future research. My specific comments are as follows:

The study relies on aggregated data from only two days, with less than half of the grids classified as valid. The discussion should acknowledge the representativeness issues, potential spatial bias, and the lack of individual trajectories, as well as how these limitations may influence the robustness of the findings.

Methodological aspects: While the application of EOF analysis is a strength, the discussion does not sufficiently address its limitations. For example, the interpretation of positive and negative anomalies may carry subjectivity; EOF analysis may be sensitive to small sample sizes, and overfitting risks should be considered. A balanced comparison with BBAC\_I, highlighting both advantages and weaknesses, would strengthen the discussion.

Results aspects The identification of evacuation routes is meaningful, but the discussion does not sufficiently connect these results to real-world observations, such as official rescue reports or documented evacuation practices. More critical reflection is needed on how well the results align—or diverge—from actual post-disaster behaviors.

Future outlook: At present, the manuscript lacks a clear vision for future research. It would benefit from explicit suggestions, such as extending analyses to longer temporal sequences, integrating multiple data sources (e.g., remote sensing, social media), combining with dynamic evacuation models, or applying the approach in real-time emergency management contexts.

I recommend major revision.

Reviewer #4: Dear Author,

1. Abstract Rewritten and Expanded: Original; focused mainly on the EOF method. Revised; now includes comparative analysis of EOF1, EOF2, and BBAC_I methods, providing clearer context and stronger conclusions.

2. Analytical Framework & Location Map: Added Figure 1 (framework) and Figure 2 (study area map) to make the methodology and spatial context clearer.

3. How the introduction is set up Better: Rearranged to first talk about BBAC_I's flaws and then talk about EOF's benefits, which makes the logic flow better.

4. Clear data processing

Explained how to deal with missing data, such as:

· How many grids were taken away (54%)

· The spatial distribution of valid and invalid grids

· Why invalid grids should not be included

5. Clearer Terms: Removed words that could mean more than one thing, like "inner join," and made the filtering method for comparing grid data more clear.

6. Clarifications about how things look and where they are:

Changed numbers to:

· Use real macroseismic intensity maps

· Make it clear that the colour of the figures has to do with the validity of the grid, not the intensity of the seismic waves.

· Combine POIs into existing maps to cut down on the number of figures.

7. Better Methodological Descriptions: Added a math explanation of BBAC_I and a way to understand what positive and negative EOF values mean.

8. Better alignment between discussion and results: moved content from the discussion to the results to stop results from leaking and removed or clarified guesses that went beyond the data scope.

9. Language and Citation Improvements: The manuscript was proofread by a native speaker, and the citation formatting was corrected to match PLOS ONE style.

10. Data Availability and Ethics Clarified:Detailed the data sharing limitations (owned by China Unicom, cannot be shared publicly) and provided contact information for data access requests.

**Do you want your identity to be public for this peer review?** For information about this choice, including consent withdrawal, please see our Privacy Policy

Reviewer #3: No

Reviewer #4: **Yes: ** Mehmet Ali Yucel

---

## [Editor Report · Acceptance letter]

PONE-D-25-08571R1

PLOS ONE

Dear Dr. Mao,

I'm pleased to inform you that your manuscript has been deemed suitable for publication in PLOS ONE. Congratulations! Your manuscript is now being handed over to our production team.

Kind regards,

on behalf of

Dr. James Colborn

Academic Editor

PLOS ONE